# Comparative statistical analysis of the release kinetics models for nanoprecipitated drug delivery systems based on poly(lactic-co-glycolic acid)

Nathaly S. Heredia[1][ᵒ], Karla Vizuete[2][ᵒ]*, Marco Flores-Calero[3], Katherine Pazmiño V.[2], Fernanda Pilaquinga[4], Brajesh Kumar[5], Alexis Debut[1,2]

1 Departamento de Ciencias de la Vida y la Agricultura, Universidad de las Fuerzas Armadas ESPE, Sangolquí, Pichincha, Ecuador, 2 Centro de Nanociencia y Nanotecnología CENCINAT, Universidad de las Fuerzas Armadas ESPE, Sangolquí, Pichincha, Ecuador, 3 Departamento de Eléctrica, Electrónica y Telecomunicaciones, Universidad de las Fuerzas Armadas ESPE, Sangolquí, Pichincha, Ecuador, 4 Escuela de Ciencias Químicas, Pontificia Universidad Católica del Ecuador, Quito, Pichincha, Ecuador, 5 Department of Chemistry, TATA College, Chaibasa, Jharkhand, India

ᵒ These authors contributed equally to this work.
* ksvizuete@espe.edu.ec

**Data Availability Statement:** All relevant data are within the paper and its Supporting information files.

## Abstract

Poly(lactic-co-glycolic acid) is one of the most used polymers for drug delivery systems (DDSs). It shows excellent biocompatibility, biodegradability, and allows spatio-temporal control of the release of a drug by altering its chemistry. In spite of this, few formulations have reached the market. To characterize and optimize the drug release process, mathematical models offer a good alternative as they allow interpreting and predicting experimental findings, saving time and money. However, there is no general model that describes all types of drug release of polymeric DDSs. This study aims to perform a statistical comparison of several mathematical models commonly used in order to find which of them best describes the drug release profile from PLGA particles synthesized by nanoprecipitation method. For this purpose, 40 datasets extracted from scientific articles published since 2016 were collected. Each set was fitted by the models: order zero to fifth order polynomials, Korsmeyer-Peppas, Weibull and Hyperbolic Tangent Function. Some data sets had few observations that do not allow to apply statistic test, thus bootstrap resampling technique was performed. Statistic evidence showed that Hyperbolic Tangent Function model is the one that best fit most of the data.

## Introduction

For more than 50 years, through nanotechnology and biomedicine, formulations and devices have been developed with the aim of improving the delivery and transport of drugs inside the human body [1–3]. The use of drug delivery systems (DDS) has brought great benefits such as altered drug pharmacokinetics, increased bioavailability, controlled and sustained drug

**Funding:** This study was financially funded by Universidad de las Fuerzas Armadas ESPE (website: www.espe.edu.ec) under the project 2020-PIC-010-CTE led by A.D. The funder had no role in study design, data collection and analysis, decision to publish, or preparation of the manuscript. There was no additional external funding received for this study.

**Competing interests:** The authors have declared that no competing interests exist.

release, target therapy, combinatorial therapy, versatility in presentation modalities, simplified treatment schedules, suit clinical needs and patient preferences, decrease side effects, and increase patient treatment compliance [4–11]. DDS generally include: microsized forms as polymeric microspheres [12], microsponges [13], microneedles [14], and nanosized devices like liposomes [15], niosomes [16], metals nanoparticles [17], magnetic nanoparticles [18], nanocrystals [19], organic nanotubes [20], exosomes [21], and polymeric nanoparticles [22]. Especially, polymeric nanomaterials have attracted attention as DDS due to their excellent biocompatibility [23, 24] and biodegradability properties [25], as well as the *spatio-temporal* control of drug delivery [26]. Another attractive feature is that they allow the encapsulation of drugs with low water solubility and/or a short in vivo life, thus increasing their bioavailability over time [27].

One of the synthetic biodegradable polymers used in DDS is Poly (lactic-co-glycolic acid) or PLGA. This is a biopolymer synthesized from the polymerization of polyglycolic acid (PGA) and polylactic acid (PLA) monomers [28]. PLGA is stable, biocompatible, degradable under physiological conditions, non-immunogenic, and non-toxic, and commercially available with a wide range of different physicochemical properties [29]. The selection of specific characteristics such as its molecular weight, inherent viscosity or the LA:GA ratio allows to control the duration and behavior of the release profile of a PLGA DDS [28, 30–32]. These particularities make it suitable in applications such as tissue engineering [33], bioimaging [34], therapeutic devices, and pharmaceutical formulations as an excipient or as part of DDSs [35–37]. Therefore, several PLGA-based products are approved by the FDA and the European Medical Agency and are marketed to treat different conditions [28, 38, 39].

PLGA-based DDSs include morphologies such as microparticles or microspheres [40], nanofibers [41], films [42], foams [43], gels [44], nanoparticles [45, 46], among others [30, 35]. Being of a size that allows them to navigate freely through the body and remain for a long period of time in the blood circulatory system, nanoparticles, especially polymeric ones, have shown great potential as DDS [47, 48]. For instance, for long-term administration formulations, the time of permanence in blood varies between 24 hours (i.e. Lupron®, AbbVie Inc. [49]), 4 weeks (Lupron Depot®, AbbVie Inc. [50]), 12 weeks (Lupaneta Pack®, AbbVie Inc. [51]), to 24 weeks (i.e. Triptodur®, Arbor Pharmaceutics [52]). To optimally develop a drug delivery system, the release mechanisms and physicochemical processes involved in the transport and release of the active ingredient within an organism must be known, especially when working with nanometric systems. The main role of PLGA in these systems is the regulation of drug release kinetics in order to achieve a sustained and controlled release. This is made possible due to the ability to modify the biodegradability of the polymer by altering its chemistry (lactide/glycolide ratio), molecular weight, stoichiometry, functionalization of the terminal groups [53, 54], and by changing its size, shape, and porosity [55]. In addition, release disturbances caused by the particle's ability to absorb water and polymer-drug and drug-drug interactions may occur [30, 38, 56–58]. Drug transport systems made with PLGA nanoparticles tend to show a biphasic behavior in which initially there is a sudden release dominated by the diffusion phenomenon in which a high concentration of active agent is rapidly released in a short period of time [59], followed by a period of slow and continuous release, where polymer degradation/erosion is the most influential mechanism [60]. The initial stage is due to the presence of non-encapsulated drug molecules on the surface, drug particles attached to the surface of the nanoparticle [61], diffusion of the active compound through the polymeric matrix, diffusion of the encapsulated agent through water-filled pores, penetration of the surrounding medium from the surface to the center of the nanoparticle and activation of the hydrolytic degradation of the PLGA [58, 62]. Initial hydrolytic degradation of the polymer and diffusion of the drug through the polymer matrix are also present, although their impact is minor [30, 40,

58]. The hydrophilic or hydrophobic nature of the encapsulating agent and its molecular weight also play an important role in this phase [32]. This initial stage can be shortened or eliminated altogether by adjusting the nanoparticle synthesis technique [63]. The second phase is due to the integration of more complex mechanisms such as the existence of different crystalline and hydrated forms of the drug, the interactions formed between polymer and drug, and the plugging of solution-filled pores, which is the result of the formation of bonds between polymer chains [32, 64]. Sometimes a third and final phase is observed, in which the release increases because the degradation of the polymer within the nanoparticle is catalyzed by the decrease in pH generated by the acidic residues produced by the degradation of the nanoparticle surface. This phenomenon is known as autocatalytic heterogeneous degradation and is frequently observed in large particles [63].

Despite the benefits of using micro- and nanoformulations in drug delivery systems, not many products have reached the market [65]. This demonstrates the difficulty of developing PLGA encapsulations and consequently their high manufacturing cost, since at each stage of the design cycle, delivery methods need to be evaluated [66]. The development of advanced DDS depends on a carefully studied and monitored selection of the components. This can be facilitated by the use of mathematical models that allow predicting the rates and behavior of the drug release process, over-time drug concentration in the body, and the biological efficacy of developed DDSs [9, 67, 68]. By choosing the right model it would be possible to reduce the number of experiments required and gain an understanding of the physicochemical dynamic of the phenomena, thus facilitating the development of new pharmaceutical products [58, 69].

Mathematical models in general are able to interpret and predict experimental findings and are used as an abstraction of a real system with assumptions and simplifications [70]. These mathematical expressions provide a qualitative and quantitative description of the main phenomena involved in drug release by incorporating more parameters in their equation, related or not to physicochemical properties [71–73].

These models can be: i) mechanistic realistic: real phenomena such as drug diffusion or dissolution, erosion, swelling, precipitation, and degradation of the polymer are taken into account [74], so they can be complex to apply, or ii) empirical/semi-empirical: not based on real chemical, physical or biological phenomena but are partially realistic from the physical-chemical point of view and are therefore generally easy to apply and explain [75].

This study aims to perform a comparative statistical analysis of existing mathematical models of drug release kinetics from PLGA nanoparticle-based DDSs engineered by nanoprecipitation to find which one could be the most suitable general model for describing the release profile.

## Materials and methods

Forty data sets describing the over-time drug release of different PLGA nanoformulations as synthesized using nanoprecipitation techniques were extracted from twenty scientific articles. The articles were selected by searching for the following keywords or combinations: "drug release systems", "poly(lactic-co-glycolic acid)", "PLGA", "nanoprecipitation" in academic search engines such as Google Scholar, PubMed y NCBI. To limit the study, the authors selected scientific articles from 2016. Furthermore, to the best of our knowledge, this was the year in which the mathematical model of hyperbolic tangent function was applied for the first time to evaluate the release profile of core-shell lipid nanoparticles [76]. From each data set, the characteristics of the nanoparticle such as encapsulated drug, size, shape, and surface modifications, as well as the characteristics of the polymer used such as LA/GA ratio and molecular weight or inherent viscosity, among others, were listed. Fig 1 illustrates the search process for

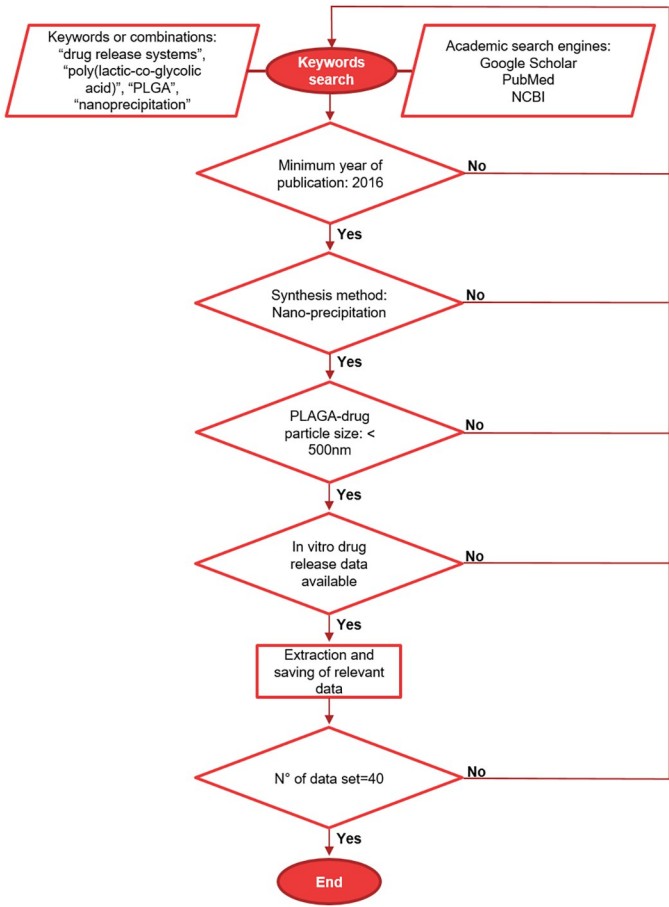

**Fig 1. Flow diagram for the screening procedure and selection criteria for the scientific articles chosen in this study.**

the scientific articles selected for this study. The data sets obtained can be found in the supplementary information.

Forty data sets obtained were fitted by linear sum-of-squares regression to five conventional empirical/semi-empirical models:

*Zero-order*: The release kinetics of an agent is only a function of time, and the process takes place at a constant rate independent of the concentration of the active agent [77].

$$Q_t = k_0 \times t + Q_0 \tag{1}$$

Where:

$Q_t$ = percentage of the released drug after time t.

$Q_0$ = initial percentage of the released drug (Usually 0).

$k_0$ = rate constant for zero-order kinetics.

$t$ = time.

*First-order*: First-order release kinetics states that the variation in concentration with respect to time depends only on the concentration of the active agent remaining in the

device [78].

$$\ln(Q_t) = \ln(Q_o) + k_1 \qquad (2)$$

Where:

$Q_t$ = percentage of the released drug after time $t$.

$Q_0$ = initial percentage of the released drug (Usually 0).

$k_1$ = rate constant for first-order kinetics.

$t$ = time.

*Korsmeyer-Peppas*: It is used to describe drug release from a polymeric system considering non-fickian mechanisms [79]. The model is useful when the release mechanism is unknown or when more than one type of drug release phenomenon is involved [77, 80]. Even if Peppas (1985) points out that this model is applicable only for the first 60% of the release curve; however, we analyzed it in both cases, 60% and 100% drug release profile [81].

$$\ln(Q_t) = \ln(k_{KP}) + n\ln(t) \qquad (3)$$

Where:

$Q_t$ = percentage of the released drug after time $t$.

$k_{KP}$ = constant of nanoparticles incorporating geometric characteristic structures.

$n$ = release exponent (related to the drug release mechanism).

$t$ = time.

*Weibull*: It is an empirical model that serves as a theoretical basis for almost all release kinetics in heterogeneous matrices. It represents a distribution function describing a phenomenon associated with a finite time [40, 80].

$$\ln[-\ln(1 - Q_t)] = \ln(\alpha) + \beta \cdot \ln(t) \qquad (4)$$

Where:

$Q_t$ = percentage of the released drug after time $t$.

$\alpha$ = scale parameter, defines the timescale of the process.

$\beta$ = curve shape factor.

$t$ = time.

*Hyperbolic tangent function*: This function arises from the modification of the diffusive release model used by Korsmeyer-Peppas with the intention of being able to extend the mathematical expression to the entire release curve of a homogeneous particle [73, 76].

$$\operatorname{atanh}(Q_t/Q_\infty) = \alpha \; t^{\frac{1}{2}} + \beta \qquad (5)$$

Where:

$Q_\infty$ = total percentage of the released drug.

$Q_t$ = percentage of the released drug after time $t$.

$\alpha$ = constant related to particular size and diffusion constant.

Fig 2 shows a time-line of the development of the mathematical models used in this study.

In addition, the 40 data sets were fitted by regression to second to fifth order polynomials, according to the equations shown in S1 File.

The fitting process of each model was carried out by means of simple linear regression analysis by sum of squares in the free software R [87]. Model acceptance criteria, such as the linear regression indexes (intercept I and slope S), regression coefficient ($R^2$), adjusted or corrected regression coefficient ($R^2_a$), sum of squares of residual (SSR), sum of squares of the error (SSE), sum square of total variation (SST), the T-statistic and the F-statistic were calculated using the function "lm" in R [88].

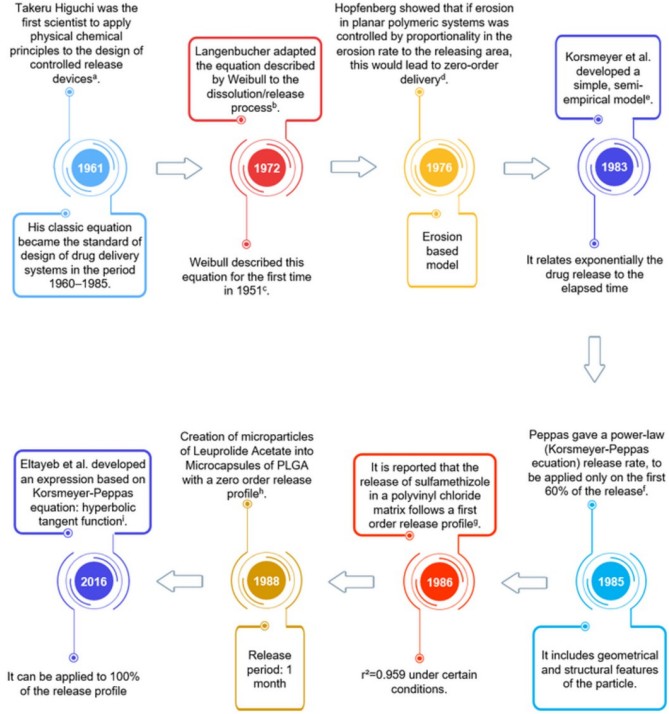

**Fig 2. Time-line of conventional empirical/semi-empirical models.** a. [82]; b. [83]; c. [84]; d. [85]; e. [79]; f. [81]; g. [86]; h. [50]; i. [76].

The significance test of the t-statistic (t-value) was used to evaluate the capacity of the parameters to linearly relate the dependent variable to the independent variable. In this test the following hypotheses were considered:

$$\text{Null hypothesis } H_0 : \beta_i = 0$$

$$\text{Alternative hypothesis } H_a : \beta_i \neq 0$$

Considering $\beta_i$ as the *i-th* analyzed parameter. Following the rejection rule, for a significance level of $\propto = 0.05$, in a two-tailed test, the critical value according to the probability table is $|t| > 1.96$, therefore:

$$\textit{Reject } H_0 \textit{ if } |t| \geq 1.96$$

$$\textit{Reject } H_0 \textit{ if } p - valor \leq 0.05$$

By rejecting $H_0$, it is concluded that $\beta_i \neq 0$ and hence a statistically significant relationship between the variables analyzed [89].

In case of the intercept, when the null hypothesis was accepted by means of the critical value or the p-value, it was recalculated considering the intercept as 0. If the null hypothesis is accepted for the slope, then the model was rejected.

On the other hand, the significance test of F-statistic, like t-statistic, indicates whether there is a significant relationship between the independent and dependent variables. In this sense, if only one parameter is analyzed, it will lead to the same conclusion as the t-statistic, but, having

more parameters, F-test can prove that there is an overall significant relationship. In this case, same hypotheses were assumed:

$$H_0 : \beta_i = 0$$

$$H_a : \beta_i \neq 0$$

The rejection criteria were:

$$Reject\ H_0\ if\ F \geq F_{\propto}$$

$$Reject\ H_0\ if\ p - value \leq\ 0.05$$

By rejecting $H_0$, it is concluded that there is a statistically significant relationship between the analyzed variables [89], conversely, if $p - value > 0.05$ the model was rejected.

Additionally, using the R function *"glance"* of the package *"broom"* the value of the Akaike Information Criterion (AIC) and Bayesian Information Criterion (BIC) were calculated for each of the fits [90].

The equations for the mathematical model acceptance criteria are summarized in S1 File.

Whereas the data sets obtained from literature contain few cumulative percentage values over time (3–14 per set), Bootstrap resampling technique was employed. This is a resampling method for statistical inference when the original sample is small. The technique seeks to create multiple data subsets from a real data set without making arbitrary assumptions, with the new data set being equally representative of the target population [91]. It is also commonly used to estimate confidence intervals, bias, and variance of an estimator or to calibrate hypothesis tests [92], see Fig 3.

In this study, the Bootstrap resampling technique was used to estimate the values of intercept, slope, and $R^2$ with 1000 Bootstrap samples using the "Boot" function. This requires the packages: "carData" [93], "car" [94], and "MUMIn" [95]. To correct the AIC value, we applied the function "boot.stepAIC" of the libraries "bootStepAIC" [96] and "MASS" [97]. In addition, the confidence interval for each of the parameters corrected to 95% was calculated with the function "confint" (S3 File).

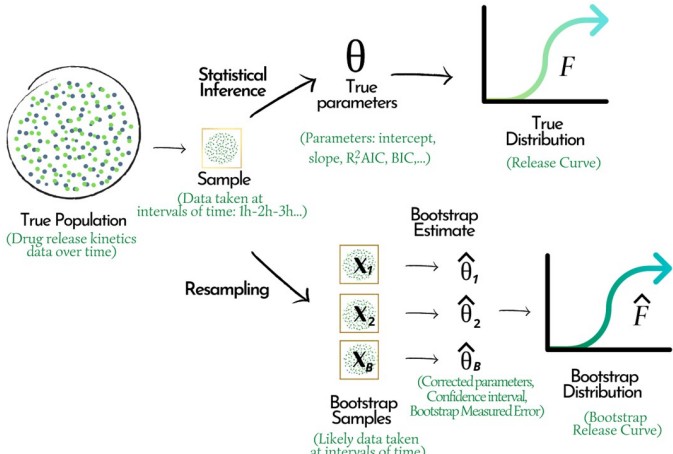

**Fig 3. Diagram of the process for parameter estimation by Bootstrap resampling.**

## Results

The cumulative drug release data extracted from the scientific articles collected are described in detail with the data in S2 File. Also, Table 1 briefly describes the polymer used for the synthesis, the drug encapsulated, the main characteristics of the nanoparticles obtained, the parameters under which the release assay was performed (release medium and duration) and the percentage of drug release achieved in each case. It can be observed that in 80% of the tests carried out, PLGA with an L/G ratio of 50:50 was used, being the polymer with a molecular weight of 7–17 kD the most commonly used. One hundred percent of the nanoparticles generated are nearly spherical and have a size range between 60 and 550 nm, and only 25% of the formulations performed surface modifications. On the other hand, the drug release conditions were variable. 32 of the release assays were performed in Phosphate buffer saline (PBS), 3 in hydrochloric acid, 2 in Phosphate buffer (PB), 2 in Fetal bovine serum (FBS) and 1 in Simulated body fluids (STF). The time to study drug release ranged from 3 hours to 21 days (504 hours). Finally, the number of observations per set was also variable. 20% of the data presented 6 observations, 17.5% 8 observations, 15% 14 observations, 12.5% 7 observations, 10% 5 observations, another 10% 9 observations, 5% 3 observations and only one data set presented 4, 10, 11 and 12 observations respectively.

S3 File shows, by means of an example using set number 33 [111] and its fit to the Korsmeyer-Peppas model, the program used to fit the mathematical models in the statistical program R.

S4 File shows the results of the fit and the acceptance criteria for each studied mathematical model and polynomial. The first criteria for rejecting a mathematical model was to evaluate the slope and F-statistic, after recalculating with intercept = 0 if necessary. Table 2 shows the results obtained. In the case of zero-order fit, sets 7–10, 18 and 19 were rejected, for first-order sets 5–11, 18, 19, 28, 29, 34 and 40, for Korsmeyer-Peppas 100% sets 7–9, 18 and 19, for Korsmeyer-Peppas 60% sets 7–10, 18, 26, 27, 28 and 36, for Weibull sets 7–9 and 18, and for Hyperbolic Tangent only the fit for sets 7 to 9 was rejected. It is important to note that no model was adequately fit for sets 7, 8 and 9. In some cases the number of observations limited the application of the statistical analysis in R. Specifically for the Korsmeyer-Peppas 60% model, it was not possible to perform the analysis on sets 7–10, 18, 26, 26, 27 and 36 because with the two first observations the set exceeded 60% of the release curve. We analyzed $R^2$- $R^2_a$ values, and AIC-BIC of previously un-rejected models. The best fit for each dataset is detailed in Table 3. According to these criteria, none of the datasets fit the zero-order and first-order models. Neither model fitted Weibull according to AIC and BIC. When studying in parallel $R^2$- $R^2_a$ and AIC-BIC only 48% of the sets accept the same model under these two criteria, which indicates that they are not equivalent. The mathematical model with the largest number of data sets with adequate fit is the hyperbolic tangent. Following $R^2$- $R^2_a$ and AIC-BIC criteria, 48% and 65% of the sets fit this model best, respectively. In addition, sets 7, 8 and 9 sets (8% of the data) did not fit any model.

Regarding the fit to the polynomial equations, the value of the intercept was analyzed (*a* letter in all polynomial functions) following the criteria mentioned in the methodology section. If null hypothesis was accepted, then the statistics were recalculated with a value of intercept = 0. If the value of the last term, represented by the letters *c*, *d*, *e*, *f* for each polynomial degree respectively, was equal to zero the model was rejected since the equation is transformed into one of a lower degree. The results are described in Table 4. Through this analysis it was found that in this particular case, the results obtained by AIC-BIC y $R^2$—$R^2_a$ are the same for all sets. Nine of the forty datasets did not fit any model. In accordance with the t-value and p-value of the last term, 11 data sets were rejected for the second-degree equation, 23 for the

**Table 1. Main characteristics of the 40 datasets analyzed.**

| No | Drug | Polymer | Lactide: Glycolide ratio | MW [kDa] or Inherent viscosity [dL g$^{-1}$] | NP shape | NP size [nm] | Surface modification | Release medium | Time range [hours] | Total Release percentage [%] | Year | Ref. |
|---|---|---|---|---|---|---|---|---|---|---|---|---|
| 1 | Paclitaxel | PLGA | (50:50) | NM | Spherical | 166.9±5.0 | - | PBS | 168 | 80 | 2016 | [98] |
| 2 | Paclitaxel | PLGA-Tween 80 | (50:50) | NM | Spherical | 156.5±8.6 | - | PBS | 168 | 75 | 2016 | [98] |
| 3 | Simvastatin | PLGA | (50:50) | 0.2 dL g$^{-1}$ | Spherical | 99–129 | TPGS Liposome | PBS | 240 | 68 | 2018 | [99] |
| 4 | Ketamine | PLGA-PEG | (50:50) | 5–55 kDa | Spherical | 98.8±0.9 | - | PBS | 504 | 82 | 2020 | [100] |
| 5 | Ketamine | PLGA-PEG | (50:50) | 5–55 kDa | Spherical | 80–300 | Shellac | PBS | 504 | 58 | 2020 | [100] |
| 6 | Carbamazepine | PLGA | (50:50) | 7–17 kDa | Spherical | 180.6±6.2 | - | PB | 48 | 90 | 2020 | [101] |
| 7 | Levetiracetam | PLGA | (50:50) | 7–17 kDa | Spherical | 180.6±6.2 | - | PB | 48 | 100 | 2020 | [101] |
| 8 | Ibuprofen | PLGA-PVA | (50:50) | 24–38 kDa | Spherical | 100–200 | - | PBS | 3 | 100 | 2017 | [102] |
| 9 | Ibuprofen | PLGA-PVA | (50:50) | 24–38 kDa | Spherical | 100–200 | - | PBS | 3 | 100 | 2017 | [102] |
| 10 | Dil | PLGA | (50:50) | 24–38 kDa | Spherical | 157.9±6.1 | - | PBS | 24 | 88 | 2017 | [103] |
| 11 | Dil | PLGA | (50:50) | 24–38 kDa | Spherical | 230.8±4.3 | - | PBS | 24 | 79 | 2017 | [103] |
| 12 | N-acetilcysteine | PLGA | (75:25) | 25–90 kDa | Spherical | 145–205 | - | PBS | 168 | 100 | 2018 | [104] |
| 13 | Dexamethasone | PLGA-PEG | (50:50) | 5 kDa | Spherical | 250–400 | - | PBS | 360 | 60 | 2017 | [105] |
| 14 | Paclitaxel | PLGA | (50:50) | 1–50 kDa | Spherical | 140±5 | - | PBS | 240 | 86 | 2017 | [106] |
| 15 | Paclitaxel | PLGA | (75:25) | NM | Spherical | 120±5 | - | PBS | 240 | 88.5 | 2017 | [106] |
| 16 | Paclitaxel | PLGA | (50:50) | 1–50 kDa | Spherical | 140±5 | - | PBS | 240 | 87.5 | 2017 | [106] |
| 17 | Paclitaxel | PLGA | (75:25) | 128 kDa | Spherical | 120±5 | - | PBS | 240 | 89 | 2017 | [106] |
| 18 | Paclitaxel | PLGA | (50:50) | NM | Spherical | 60–450 | - | FBS | 3 | 100 | 2017 | [107] |
| 19 | Sorafenib | PLGA | (50:50) | NM | Spherical | 70–550 | - | FBS | 3 | 100 | 2017 | [107] |
| 20 | Ropinirole HCl | PLGA-PVA | (50:50) | NM | Spherical | 308.3±23 | Polysorbate 80 | PBS | 24 | 42 | 2020 | [108] |
| 21 | Ropinirole HCl | PLGA-PVA | (50:50) | NM | Spherical | 345.5±42 | Polysorbate 80 | PBS | 24 | 36 | 2020 | [108] |
| 22 | Ropinirole HCl | PLGA-PVA | (50:50) | NM | Spherical | 264.6±11 | Polysorbate 80 | PBS | 24 | 43 | 2020 | [108] |
| 23 | Ropinirole HCl | PLGA-PVA | (50:50) | NM | Spherical | 212±3 | Polysorbate 80 | PBS | 24 | 58 | 2020 | [108] |
| 24 | Ropinirole HCl | PLGA-PVA | (50:50) | NM | Spherical | 351.9±20 | Polysorbate 80 | PBS | 24 | 44 | 2020 | [108] |
| 25 | Ropinirole HCl | PLGA-PVA | (50:50) | NM | Spherical | 196.4±8 | Polysorbate 80 | PBS | 24 | 62 | 2020 | [108] |
| 26 | Ciprofloxacin | PLGA-PEG | (50:50) | 30–35 kDa | Spherical | 174–205 | - | STF | 144 | 95 | 2019 | [109] |
| 27 | Methotrexate | PLGA-PVA | (50:50) | 7–17 kDa | Spherical | 78.1±7.3 | - | PBS | 72 | 92 | 2019 | [46] |
| 28 | Methotrexate & Curcumin | PLGA-PVA | (50:50) | 7–17 kDa | Spherical | 148.3±4.1 | - | PBS | 72 | 80 | 2019 | [46] |
| 29 | Curcumin | PLGA-PVA | (50:50) | 7–17 kDa | Spherical | 92.4±5.2 | - | PBS | 72 | 75 | 2019 | [46] |
| 30 | Methotrexate & Curcumin | PLGA-PVA | (50:50) | 7–17 kDa | Spherical | 148.3±4.1 | - | PBS | 72 | 57 | 2019 | [46] |
| 31 | Curcumin | PLGA | (75:25) | 65 kDa | Spherical | 280±16 | - | PBS | 120 | 68 | 2018 | [110] |
| 32 | Curcumin | PLGA | (75:25) | 65kDa | Spherical | 251±3 | Anti-EGFRvIII antibodies | PBS | 120 | 69 | 2018 | [110] |
| 33 | MTPC | PLGA-PEG | (50:50) | 17 kDa | Spherical | 143.2±3.7 | - | PBS | 4 | 62 | 2020 | [111] |
| 34 | Lansoprazole | PLGA | (50:50) | NM | Spherical | 246.7±3.4 | - | PBS | 48 | 92 | 2019 | [112] |
| 35 | Saquinavir | PLGA | (50:50) | NM | Spherical | 412.2 | - | PBS | 16 | 99 | 2020 | [113] |
| 36 | Insulin | PLGA-PEG | (50:50) | 10 kDa | Spherical | 78.7 | Zinc chloride | PBS | 8 | 100 | 2017 | [37] |
| 37 | Paclitaxel | PLGA-PEG | (50:50) | 55 kDa | Spherical | 100 | - | PBS | 168 | 39 | 2020 | [114] |
| 38 | Olmesartan medoxomil | PLGA | (75:25) | 4–15 kDa | Spherical | 71.2±4.5 | - | HCl | 24 | 39 | 2016 | [45] |
| 39 | Olmesartan medoxomil | PLGA | (75:25) | 4–15 kDa | Spherical | 78.9±3.2 | - | HCl | 24 | 30 | 2016 | [45] |

*(Continued)*

**Table 1.** (Continued)

| No | Drug | Polymer | Lactide: Glycolide ratio | MW [kDa] or Inherent viscosity [dL g$^{-1}$] | NP shape | NP size [nm] | Surface modification | Release medium | Time range [hours] | Total Release percentage [%] | Year | Ref. |
|---|---|---|---|---|---|---|---|---|---|---|---|---|
| **40** | Olmesartan medoxomil | PLGA | (75:25) | 4–15 kDa | Spherical | 97.8±6.3 | - | HCl | 24 | 35 | 2016 | [45] |

Notation: MW: molecular weight, NM: Not mentioned in the source, "-": Not applicable, PLGA: poly (lactic-co-glycolic acid), PVA: polyvinyl alcohol; PEG: polyethylene glycol, TPGS; Dil: 1,10-dioctadecyl-3,3,30,30-tetramethylindocarbocyanine perchlorate, MTPC: N′-(2-Methoxybenzylidene)-3-methyl-1-phenyl-H-Thieno [2,3-c]Pyrazole-5-arbohyd-razide, PBS: Phosphate buffer saline, PB: Phosphate buffer, STF: Simulated body fluids, FBS: Fetal bovine serum, HCl: hydrochloric acid.

third-order equation, 27 for the fourth-order equation, and 25 for the fifth-order equation. AIC-BIC and $R^2$—$R^2_a$ showed that the polynomial equation that provided the best fit to the largest amount of data was the second-degree equation (38%), followed by the third-degree equation (20%), the fifth-degree equation (13%) and finally the fourth-degree equation (8%). It is worth noting that when the average $R^2$—$R^2_a$ of the data sets unrejected previously were evaluated, it was found that these values tend to increase as the degree of the polynomial increases, as can be seen in Fig 4.

Moreover, when Bootstrap resampling was performed (S5 File) it could be found that the best fit for data sets 7, 8 and 9 is Korsmeyer-Peppas 100% or zero-order by $R^2$ and Korsmeyer-Peppas 100% by AIC, see Table 5. More specifically, by AIC, 30% of the sets present a better fit with Korsmeyer-Peppas 100% and 70% with Hyperbolic Tangent for polynomial equations 13%, 23%, 23%, 23%, and 33% were fit to second to fifth degree polynomials respectively. In addition, a single data set fitted to all polynomials and 3 to none. It is important to note that by analyzing the coefficient of determination of the resampled data, more than one model is accepted. As the best fit, by using the AIC with the most negative value, the model or polynomial equation that best fits the data can be determined.

Finally, Figs 5–7 show scatter diagrams of $R^2$ in relation to AIC and BIC for the models analyzed, both for the values obtained by linear regression and the Bootstrap corrected values. Additionally, the region of what is considered a good fit is indicated: between 0.8 and 1 for $R^2$ and between 0 and -inf for AIC and BIC. In both figures it can be clearly observed how in the Hyperbolic Tangent Function model the data sets are mostly grouped in the region considered as having the best fit.

## Discussion

In this study, five general controlled drug release models and four polynomial equations, from the second to the fifth degree, were statistically analyzed. The main objective was to find a mathematical expression that can describe the release mechanism of an active compound observed in PLGA nanoparticles synthesized by nanoprecipitation method.

While it is true that mathematical models are tools that help to understand the dynamics of a drug release, and to save time and money [115], it is important to note that these also have shortcomings in terms of predictive power and in explaining and addressing all the physical, chemical and biological phenomena involved since they mainly describe the dominant event [62]. Among them is the initial burst release, as the fraction released can sometimes approach 80% of the total encapsulated agent [116] in a time that may be less than 10% of the duration of the entire release process [48, 52]. In addition, very few models described in the literature

**Table 2. Evaluation of the p-value of the intercept, slope and F-statistic of the datasets.**

| Model is rejected if: | Zero order | | | First order | | | Korsmeyer Peppas 100% | | | Korsmeyer Peppas 60% | | | Weibull | | | Hyperbolic Tangent | | |
|---|---|---|---|---|---|---|---|---|---|---|---|---|---|---|---|---|---|---|
| | $\lvert t\rvert<1{,}96$ $p>0{,}05$ I | $\lvert t\rvert<1{,}96$ $p>0{,}05$ S | $p>0{,}05$ Est F | $\lvert t\rvert<1{,}96$ $p>0{,}05$ I | $\lvert t\rvert<1{,}96$ $p>0{,}05$ S | $p>0{,}05$ Est F | $\lvert t\rvert<1{,}96$ $p>0{,}05$ I | $\lvert t\rvert<1{,}96$ $p>0{,}05$ S | $p>0{,}05$ Est F | $\lvert t\rvert<1{,}96$ $p>0{,}05$ I | $\lvert t\rvert<1{,}96$ $p>0{,}05$ S | $p>0{,}05$ Est F | $\lvert t\rvert<1{,}96$ $p>0{,}05$ I | $\lvert t\rvert<1{,}96$ $p>0{,}05$ S | $p>0{,}05$ Est F | $\lvert t\rvert<1{,}96$ $p>0{,}05$ I | $\lvert t\rvert<1{,}96$ $p>0{,}05$ S | $p>0{,}05$ Est F |
| 1 | A | A | A | A | X | X | A | A | A | A | A | A | A | A | A | A | A | A |
| 2 | A | A | A | A | X | X | A | A | A | A | A | A | A | RC | A | A | A | A |
| 3 | A | A | A | A | A | A | A | A | A | A | A | A | A | A | A | A | A | A |
| 4 | A | A | A | A | A | A | A | A | A | RC | A | A | A | A | A | A | A | A |
| 5 | A | A | A | A | X | X | RC | A | A | RC | A | A | A | A | A | RC | A | A |
| 6 | A | A | A | A | X | X | A | A | A | A | A | A | A | A | A | RC | A | A |
| 7 | A | X | X | RC | X | X | A | X | X | Insufficient data | | | Insufficient data | | | Insufficient data | | |
| 8 | RC | X | X | RC | X | X | A | X | X | | | | | | | | | |
| 9 | RC | X | A | A | X | X | A | X | X | | | | | | | | | |
| 10 | RC | A | X | A | X | X | A | A | A | A | A | A | RC | A | A | RC | A | A |
| 11 | RC | | | A | X | X | A | A | A | A | A | A | A | RC | A | A | A | A |
| 12 | A | A | A | A | A | A | A | A | A | A | A | A | A | A | A | A | A | A |
| 13 | A | A | A | A | A | A | A | A | A | A | A | A | A | A | A | A | A | A |
| 14 | A | A | A | A | A | A | A | A | A | A | A | A | A | A | A | A | A | A |
| 15 | A | A | A | A | X | X | X | X | X | X | X | X | X | X | X | A | A | A |
| 16 | A | A | A | A | A | A | A | A | A | A | A | A | A | A | A | A | A | A |
| 17 | A | A | A | A | A | A | A | A | A | A | A | A | A | A | A | A | A | A |
| 18 | RC | X | X | A | X | X | A | X | X | Insufficient data | | | RC | X | X | RC | A | A |
| 19 | RC | X | X | A | X | X | A | X | X | RC | A | A | RC | A | A | A | A | A |
| 20 | A | A | A | A | A | A | A | A | A | A | A | A | RC | A | A | A | A | A |
| 21 | A | A | A | A | A | A | A | A | A | A | A | A | A | A | A | A | A | A |
| 22 | A | A | A | A | A | A | A | A | A | A | A | A | A | A | A | A | A | A |
| 23 | A | A | A | A | A | A | A | A | A | A | A | A | A | A | A | A | A | A |
| 24 | A | A | A | A | X | X | A | X | X | A | X | X | A | A | A | A | A | A |
| 25 | A | A | A | A | A | A | A | A | A | A | A | A | A | A | A | A | A | A |
| 26 | A | A | A | A | X | X | A | A | A | Insufficient data | | | A | A | A | A | A | A |
| 27 | A | A | A | A | A | A | A | A | A | Insufficient data | | | A | A | A | A | A | A |
| 28 | A | A | A | A | X | X | A | A | A | A | X | X | A | A | A | A | A | A |
| 29 | A | A | A | A | X | X | A | A | A | A | A | A | A | A | A | A | A | A |
| 30 | A | A | A | A | A | A | A | A | A | A | A | A | A | A | A | A | A | A |
| 31 | A | A | A | A | A | A | A | A | A | A | A | A | A | A | A | A | A | A |
| 32 | A | A | A | A | A | A | A | A | A | A | A | A | A | A | A | A | A | A |
| 33 | RC | A | A | A | A | A | A | A | A | A | A | A | A | A | A | A | A | A |
| 34 | A | A | A | A | A | X | A | A | A | A | A | A | A | A | A | A | A | A |
| 35 | A | A | A | A | A | A | A | A | A | A | A | A | A | A | A | A | A | A |

*(Continued)*

**Table 2.** (Continued)

| Model is rejected if: | Zero order | | | First order | | | Korsmeyer Peppas 100% | | | Korsmeyer Peppas 60% | | | Weibull | | | Hyperbolic Tangent | | |
| --- | --- | --- | --- | --- | --- | --- | --- | --- | --- | --- | --- | --- | --- | --- | --- | --- | --- | --- |
| | I (\|t\|<1,96 p>0,05) | S (\|t\|<1,96 p>0,05) | Est F (p>0,05) | I (\|t\|<1,96 p>0,05) | S (\|t\|<1,96 p>0,05) | Est F (p>0,05) | I (\|t\|<1,96 p>0,05) | S (\|t\|<1,96 p>0,05) | Est F (p>0,05) | I (\|t\|<1,96 p>0,05) | S (\|t\|<1,96 p>0,05) | Est F (p>0,05) | I (\|t\|<1,96 p>0,05) | S (\|t\|<1,96 p>0,05) | Est F (p>0,05) | I (\|t\|<1,96 p>0,05) | S (\|t\|<1,96 p>0,05) | Est F (p>0,05) |
| 36 | A | A | A | A | A | A | A | A | A | Insufficient data | | | A | A | A | A | A | A |
| 37 | A | A | A | A | A | A | A | A | A | A | A | A | A | A | A | A | A | A |
| 38 | RC | A | A | A | A | A | A | A | A | A | A | A | A | A | A | A | A | A |
| 39 | RC | A | A | A | A | A | A | A | A | A | A | A | A | A | A | A | A | A |
| 40 | RC | A | A | A | A | X | A | A | A | A | A | A | A | A | A | A | A | A |

Notation: "I": Intercept; "S": Slope "RC": Recalculated; "A": Accepted model; "X": Rejected model.

**Table 3. Best fit for each data set according to $R^2$, $R^2_a$, AIC and BIC.**

| No. | $R^2$—$R^2_a$ | AIC—BIC | No. | $R^2$—$R^2_a$ | AIC—BIC |
|---|---|---|---|---|---|
| 1 | Korsmeyer-Peppas 60% | Korsmeyer-Peppas 60% | 21 | Hyperbolic Tangent | Hyperbolic Tangent |
| 2 | Korsmeyer-Peppas 60% | Hyperbolic Tangent | 22 | Hyperbolic Tangent | Hyperbolic Tangent |
| 3 | Weibull | Hyperbolic Tangent | 23 | Hyperbolic Tangent | Hyperbolic Tangent |
| 4 | Hyperbolic Tangent | Hyperbolic Tangent | 24 | Weibull | Hyperbolic Tangent |
| 5 | Hyperbolic Tangent | Hyperbolic Tangent | 25 | Hyperbolic Tangent | Hyperbolic Tangent |
| 6 | Hyperbolic Tangent | Hyperbolic Tangent | 26 | Hyperbolic Tangent | Korsmeyer-Peppas 100% |
| 7 | No model | No model | 27 | Weibull | Korsmeyer-Peppas 100% |
| 8 | No model | No model | 28 | Hyperbolic Tangent | Hyperbolic Tangent |
| 9 | No model | No model | 29 | Weibull | Hyperbolic Tangent |
| 10 | Hyperbolic Tangent | Hyperbolic Tangent | 30 | Weibull | Hyperbolic Tangent |
| 11 | Hyperbolic Tangent | Hyperbolic Tangent | 31 | Weibull | Hyperbolic Tangent |
| 12 | Weibull | Hyperbolic Tangent | 32 | Weibull | Hyperbolic Tangent |
| 13 | Weibull | Hyperbolic Tangent | 33 | Hyperbolic Tangent | Hyperbolic Tangent |
| 14 | Weibull | Hyperbolic Tangent | 34 | Korsmeyer-Peppas 60% | Korsmeyer-Peppas 60% |
| 15 | Weibull | Korsmeyer-Peppas 100% | 35 | Korsmeyer-Peppas 60% | Korsmeyer-Peppas 100% |
| 16 | Korsmeyer-Peppas 60% | Korsmeyer-Peppas 100% | 36 | Weibull | Korsmeyer-Peppas 100% |
| 17 | Korsmeyer-Peppas 60% | Korsmeyer-Peppas 60% | 37 | Hyperbolic Tangent | Hyperbolic Tangent |
| 18 | Hyperbolic Tangent | Hyperbolic Tangent | 38 | Hyperbolic Tangent | Hyperbolic Tangent |
| 19 | Korsmeyer-Peppas 60% | Korsmeyer-Peppas 60% | 39 | Hyperbolic Tangent | Hyperbolic Tangent |
| 20 | Hyperbolic Tangent | Hyperbolic Tangent | 40 | Hyperbolic Tangent | Hyperbolic Tangent |

consider the effect of drug properties, especially acids, on the degradation rate of polymer particles [117]. It should not be forgotten that mathematical models are approximations of reality where assumptions are made, so they should always be accompanied by experimental data that can support these assertions [118–120]. Empirical and semi-empirical mathematical models tend to have limited predictive power, so it is recommended that these theories be applied, and be of great use, when you want to compare release profiles using a specific parameter [121].

On the other hand, the coefficient of determination "$R^2$" is one of the most widely used criteria to determine whether or not a model is a better fit to a release curve. However, it has been widely demonstrated that $R^2$ can increase, i.e., tends more to one, if more variables are added to the equation, even if they are completely irrelevant to the model [122]. This problem is more accentuated when the number of observations is insufficient [123]. Similarly, $R^2$ is not considered an appropriate goodness-of-fit criterion when analyzing multi-parameter models, as our study. Therefore, taking into account the adjusted coefficient of determination $R^2_a$ is the most appropriate, although in a limited way, since its value only increases when a relevant variable is added to the model, otherwise it decreases [124]. Since there is no perfect correlation between the variables, special care should be taken in the interpretation of the results when $R^2$ or $R^2_a$ take the value of 1 [125]. In addition, it is recommended to accompany these criteria with statistical procedures such as t-test, F-test, and prediction intervals to know the quality and influence of individual observations in the estimation [126].

Since the Bayesian and Akaike criteria are based on the addition of statistical errors corrected by a penalty function proportional to the number of parameters, they can be a good alternative to determine which model offers the best fit. In addition, these criteria help to determine whether the inclusion of a term, or increased complexity in the equation, results in better fit [127]. It is important to emphasize that models with more negative values of AIC and

**Table 4. Evaluation of the p-value of the intercept, last term, and F-statistic of the data sets.**

| Model is rejected if: | 2nd order | | | 3rd order | | | 4th order | | | 5th order | | |
|---|---|---|---|---|---|---|---|---|---|---|---|---|
| | $|t|<1{,}96$ | $|t|<1{,}96$ | $p>0{,}05$ | $|t|<1{,}96$ | $|t|<1{,}96$ | $p>0{,}05$ | $|t|<1{,}96$ | $|t|<1{,}96$ | $p>0{,}05$ | $|t|<1{,}96$ | $|t|<1{,}96$ | $p>0{,}05$ |
| | $p>0{,}05$ a | $p>0{,}05$ LT | Est F | $p>0{,}05$ a | $p>0{,}05$ LT | Est F | $p>0{,}05$ a | $p>0{,}05$ LT | Est F | $p>0{,}05$ a | $p>0{,}05$ LT | Est F |
| 1 | A | X | - | A | X | - | A | X | - | RC | A | A |
| 2 | A | X | - | A | X | - | A | X | - | A | X | - |
| 3 | A | A | A | A | A | A | A | A | A | A | A | A |
| 4 | RC | A | A | RC | X | - | Insufficient data | | | Insufficient data | | |
| 5 | RC | A | A | RC | X | - | | | | | | |
| 6 | A | A | A | RC | X | - | RC | X | - | RC | X | - |
| 7 | Insufficient data | | | Insufficient data | | | Insufficient data | | | Insufficient data | | |
| 8 | | | | | | | | | | | | |
| 9 | | | | | | | | | | | | |
| 10 | RC | A | A | A | A | A | | | | | | |
| 11 | RC | A | A | RC | X | - | | | | | | |
| 12 | A | X | - | A | X | - | A | X | - | A | X | - |
| 13 | A | A | A | A | A | A | A | X | - | A | X | - |
| 14 | A | A | A | A | A | A | A | X | - | A | X | - |
| 15 | A | A | A | A | X | - | A | X | - | A | X | - |
| 16 | A | A | A | A | A | A | A | X | - | A | A | A |
| 17 | A | A | A | A | X | - | A | A | A | A | X | - |
| 18 | RC | A | A | RC | X | - | RC | X | X | Insufficient data | | |
| 19 | RC | A | A | A | A | A | A | X | - | RC | A | A |
| 20 | A | A | A | A | A | A | A | X | - | A | X | - |
| 21 | A | A | A | A | A | A | A | X | - | A | X | - |
| 22 | A | A | A | A | X | - | A | X | - | A | A | A |
| 23 | A | A | A | A | X | - | A | X | - | A | X | - |
| 24 | A | A | A | A | X | - | A | X | - | A | X | - |
| 25 | A | A | A | A | X | - | A | X | - | A | X | - |
| 26 | A | X | - | A | A | A | A | A | A | A | X | - |
| 27 | A | X | X | A | X | X | A | X | X | Insufficient data | | |
| 28 | A | X | - | A | A | A | RC | X | - | | | |
| 29 | A | X | - | RC | X | - | RC | A | A | | | |
| 30 | A | X | - | A | A | A | A | X | - | | | |
| 31 | A | A | A | A | X | - | A | X | - | A | X | - |
| 32 | A | A | A | A | X | - | A | X | - | A | X | - |
| 33 | RC | X | - | RC | X | - | RC | X | - | Insufficient data | | |
| 34 | A | A | A | A | A | A | RC | A | A | RC | X | - |
| 35 | A | A | A | A | X | - | A | X | - | RC | X | - |
| 36 | RC | A | A | RC | A | A | RC | X | - | Insufficient data | | |
| 37 | RC | A | A | Insufficient data | | | Insufficient data | | | | | |
| 38 | RC | A | A | RC | X | - | RC | X | - | RC | X | - |
| 39 | RC | X | - | RC | X | - | A | X | - | RC | X | - |
| 40 | RC | X | - | RC | X | - | RC | X | - | RC | X | - |

Notation: "LT": Last term of the polynomial equation; "RC": Recalculated; "A": Accepted model; "X": Rejected model; "-": Rejected by the previous criterion.

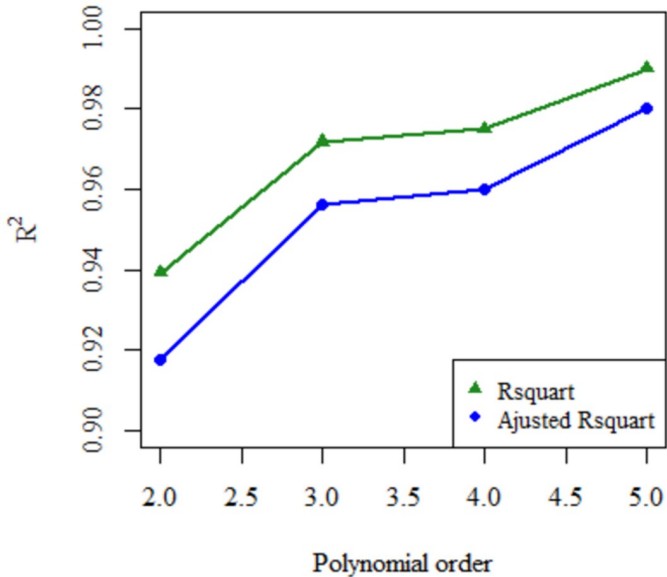

**Fig 4. Average of $R^2$—$R^2_a$ of the data sets fitted to the second to fifth order polynomials.**

BIC are likely to minimize information loss, which translates into a more appropriate choice of the model with the best fit [128].

The zero-order model is usually employed to assume a single-phase ideal condition in which the release kinetics is continuous and can be explained as a function dependent on time and independent of the dissolved concentration in the release medium [48]. In addition, zero-order behavior can be observed in materials capable of swelling [129], yet due to complicated bulk erosion and surface erosion and diffusion processes, most studies on the sustained release of hydrophilic drugs have failed to achieve a release profile that conforms to zero-order kinetics for a prolonged time [130]. This conclusion was verified in this study, in which the data sets had a biphasic behavior far from zero-order according to the low values of $R^2$ and $R^2_a$ and the high averages of the calculated AIC and BIC [131]. Likewise, the results obtained showed rejection of the first-order model for all data sets, demonstrating that the data sets do not follow a model that is dependent on the initial drug concentration [78, 132].

The Korsmeyer-Peppas equation also called "power law" is a semi-empirical model based on the diffusion phenomenon [133] that is used to describe in a general way the main transport phenomena involved in the release, either by diffusion or swelling [134]. Although a good fit has been achieved with this model, several authors have shown that Korsmeyer-Peppas is applicable only to the first 60% of the release profile and is not an appropriate equation to describe the complete release profile [73, 76, 134]. By examining only the first 60% of the data sets, some problems could be detected, such as the small number of observations in some sets, which made it difficult and even impossible to fit the data to the models [135]. It was found that 4 data sets presented a more negative AIC value when 60% of the curve was analyzed, and 6 when 100% was analyzed. This demonstrates that although the model is more complex than the previous ones, this does not result in a better description of the release process.

The empirical Weibull model is an expression that tries to find a linear relationship between the logarithm of drug release and the logarithm of time [136]. Despite not having a kinetic nature, nor parameters physically related to the release phenomena, the ability of the

**Table 5. Comparison of $R^2$ and AIC criteria of data sets using Bootstrap resampling of empirical/semiempirical models and polynomial equations.**

| No. | Zero order | | First order | | Korsmeyer-Peppas 100% | | Korsmeyer-Peppas 60% | | Weibull | | Hyperbolic Tangent | | 2nd order | | 3rd order | | 4th order | | 5th order | |
|---|---|---|---|---|---|---|---|---|---|---|---|---|---|---|---|---|---|---|---|---|
| | $R^2$ | AIC | $R^2$ | AIC | $R^2$ | AIC | $R^2$ | AIC | $R^2$ | AIC | $R^2$ | AIC | $R^2$ | AIC | $R^2$ | AIC | $R^2$ | AIC | $R^2$ | AIC |
| 1 | 0.68 | 52.37 | 0.36 | -13.25 | 0.87 | -22.41 | 0.97 | -15.19 | 0.91 | -19.73 | 0.87 | -30.68 | 0.68 | 51.5 | 0.88 | 49.26 | 0.68 | 52.37 | 0.99 | 43.16 |
| 2 | 0.68 | 49.29 | 0.4 | -16.81 | 0.87 | -26.1 | 0.93 | -17.39 | 0.91 | -23.15 | 0.85 | -33.52 | 0.68 | 48.32 | 0.89 | 45.61 | 0.68 | 49.29 | 1 | 34.25 |
| 3 | 0.8 | 57.58 | 0.54 | -23.17 | 0.98 | -55.41 | 0.97 | -52.27 | 0.99 | -58.6 | 0.96 | -67.24 | 0.8 | 47.17 | 0.95 | 41.68 | 0.97 | 34.76 | 0.97 | 34.76 |
| 4 | 0.94 | 23.08 | 0.85 | -11.21 | 0.98 | -15.09 | 0.95 | -16.98 | 0.99 | -20.95 | 0.9 | -23.3 | 0.97 | 16.52 | 1 | 15.91 | Insufficient data | | Insufficient data | |
| 5 | 0.9 | 22.02 | 0.79 | -11.23 | 0.88 | -14.28 | 0.88 | -14.28 | 0.96 | -15.43 | 0.98 | -23.3 | 0.94 | 16.73 | 1 | 18.36 | | | | |
| 6 | 0.85 | 41.55 | 0.57 | -4.86 | 0.96 | -19.35 | 0.97 | -16.73 | 0.99 | -25.41 | 0.99 | -40.32 | 0.91 | 40.17 | 0.72 | 35.17 | | | | |
| 7 | 0.81 | 11.62 | 1 | -14.96 | 0.81 | -18.13 | Insufficient data | | Insufficient data | | | | | | | | Insufficient data | | Insufficient data | |
| 8 | 0.65 | 20.41 | 1 | -3.8 | 0.65 | -5.61 | | | | | | | | | | | | | | |
| 9 | 0.79 | 19.35 | 1 | -3.45 | 0.79 | -5.71 | | | | | | | | | | | | | | |
| 10 | 0.74 | 31.81 | 0.6 | -2.4 | 0.92 | -8.9 | | | 0.96 | -9.23 | 0.98 | -19.73 | 0.99 | 25.93 | 1 | -12.11 | | | | |
| 11 | 0.9 | 28.02 | 0.72 | -3.73 | 0.97 | -13.21 | 0.99 | -12.99 | 0.98 | -14.29 | 0.92 | -23.07 | 0.99 | 19.53 | 1 | 16.45 | | | | |
| 12 | 0.88 | 55.86 | 0.59 | -14.24 | 0.94 | -28.71 | 0.83 | -13.82 | 0.96 | -22.1 | 0.96 | -34.88 | 0.88 | 55.86 | 0.88 | 55.86 | 0.95 | 53.42 | 0.99 | 55.05 |
| 13 | 0.84 | 51.58 | 0.61 | -20.64 | 0.98 | -46.53 | 0.98 | -46.53 | 0.98 | -44.79 | 0.95 | -58.51 | 0.84 | 36.89 | 0.98 | 23.08 | 0.84 | 51.58 | 0.99 | 24.32 |
| 14 | 0.78 | 43.26 | 0.59 | -19.31 | 0.97 | -37.38 | 0.97 | -20.61 | 0.99 | 37.02 | 0.96 | -34.36 | 0.78 | 35.05 | 0.78 | 35.05 | 0.78 | 43.26 | 0.78 | 43.26 |
| 15 | 0.73 | 43.82 | 0.52 | -20.38 | 0.95 | -35.01 | 1 | -15.4 | 0.98 | -33.11 | 0.94 | -30.69 | 0.73 | 43.82 | 0.73 | 43.82 | 0.73 | 43.82 | 1 | 17.25 |
| 16 | 0.81 | 38.89 | 0.63 | -24.77 | 0.97 | -43.47 | 1 | -21.1 | 0.99 | -41.73 | 0.97 | -40.7 | 0.81 | 38.89 | 0.81 | 38.89 | 1 | 28.45 | 1 | 19.47 |
| 17 | 0.74 | 39.87 | 0.55 | -26.04 | 0.96 | -41.2 | 1 | -26.82 | 0.98 | -38.34 | 0.94 | -33.19 | 0.74 | 35.13 | 0.96 | 32.44 | 1 | 22.35 | 1 | 9.29 |
| 18 | 0.62 | 43.12 | 0.34 | 2.89 | 0.6 | 2.19 | Insufficient data | | 0.47 | 3.38 | 0.82 | -3.66 | 0.54 | 40.22 | 0.78 | 42.18 | 1 | 40.35 | Insufficient data | |
| 19 | 0.68 | 49.6 | 0.32 | 4.16 | 0.63 | 2.7 | 1 | -2.91 | 0.57 | 3.32 | 0.86 | -4.56 | 0.55 | 46.33 | 0.97 | 28.38 | 1 | 22.12 | 1 | 12.73 |
| 20 | 0.88 | 48.86 | 0.65 | -28.16 | 0.98 | -64.15 | 0.98 | -64.15 | 0.98 | -62.8 | 0.99 | -100.1 | 0.88 | 10.63 | 0.99 | -1.9 | 0.79 | 48.86 | 0.99 | 1.79 |
| 21 | 0.94 | 30.67 | 0.78 | -4.34 | 0.98 | -69.71 | 0.98 | -69.71 | 0.98 | -64.97 | 0.99 | -121.2 | 0.94 | -30.87 | 0.99 | -34.67 | 0.94 | 30.67 | 0.94 | 30.67 |
| 22 | 0.92 | 40.25 | 0.73 | -38.65 | 0.98 | -66.99 | 0.98 | -66.99 | 0.98 | -61.11 | 0.99 | -104.6 | 0.92 | 7.98 | 0.92 | 40.25 | 0.99 | 8.37 | 0.99 | -2.97 |
| 23 | 0.93 | 45.66 | 0.77 | -43.02 | 0.98 | -74.29 | 0.98 | -74.29 | 0.98 | -64.26 | 0.98 | -103.8 | 0.93 | -2.83 | 0.93 | 45.66 | 0.99 | -3.81 | 0.93 | 45.66 |
| 24 | 0.9 | 42.3 | 0.71 | -38.52 | 0.99 | -81.26 | 0.99 | -81.26 | 0.99 | -79.78 | 0.98 | -109.2 | 0.9 | 12.9 | 0.9 | 12.9 | 0.9 | 12.9 | 0.9 | 12.9 |
| 25 | 0.95 | 43.41 | 0.79 | -41.53 | 0.98 | -74.89 | 0.98 | -74.89 | 0.98 | -63.79 | 0.98 | -107.4 | 0.95 | 3.08 | 0.95 | 43.41 | 0.95 | 43.41 | 0.95 | 43.41 |
| 26 | 0.92 | 27.36 | 0.79 | -46.52 | 0.95 | -54.72 | Insufficient data | | 0.92 | -32.7 | 0.98 | -38.03 | 0.92 | 27.36 | 0.92 | 27.36 | 0.92 | 27.36 | 1 | 8.18 |
| 27 | 0.85 | 27.68 | 0.7 | -21.56 | 0.96 | -31.3 | | | 0.98 | -27.57 | 0.98 | -26.98 | 0.85 | 26.4 | 0.88 | 21.39 | 1 | 5.42 | Insufficient data | |
| 28 | 0.85 | 32.68 | 0.66 | -11.72 | 0.96 | -21.91 | 1 | -16.76 | 0.97 | -20.11 | 0.97 | -23.01 | 0.85 | 27.86 | 0.99 | 10.53 | 1 | 22.83 | | |
| 29 | 0.86 | 29.37 | 0.67 | -13.71 | 0.95 | -22.55 | 0.98 | -15.33 | 0.97 | -22.12 | 0.95 | -29.74 | 0.86 | 27.44 | 0.95 | 32.93 | 1 | 21.04 | | |
| 30 | 0.9 | 24.04 | 0.72 | -16.39 | 0.98 | -29.01 | 0.98 | -29.01 | 0.98 | -28.44 | 0.96 | -35.77 | 0.9 | 21.42 | 0.9 | 24.04 | 1 | -5.4 | | |
| 31 | 0.86 | 33.03 | 0.69 | -26.95 | 0.97 | -40.84 | 0.94 | -23.68 | 0.98 | -38.61 | 0.95 | -44.25 | 0.86 | 33.03 | 0.86 | 33.03 | 1 | 26.09 | 1 | 24.84 |
| 32 | 0.84 | 34.88 | 0.66 | -24.77 | 0.96 | -37.54 | 0.93 | -16.67 | 0.97 | -35.32 | 0.94 | -41.95 | 0.84 | 34.88 | 0.84 | 34.88 | 1 | 27.35 | 1 | 26.26 |
| 33 | 0.93 | 22.81 | 0.8 | -9.06 | 0.97 | -19.02 | 0.97 | -19.02 | 0.98 | -20.4 | 0.99 | -41.9 | 0.93 | 22.81 | 0.99 | 17.67 | 1 | 8.1 | Insufficient data | |
| 34 | 0.66 | 55.04 | 0.41 | -15.52 | 0.87 | -24.93 | 0.99 | -22.07 | 0.92 | -20.17 | 0.93 | -22.17 | 0.92 | -20.17 | 0.97 | 46.08 | 0.98 | 28.11 | 0.99 | -30.1 |
| 35 | 0.97 | 31.63 | 0.87 | -25.13 | 0.99 | -56.11 | 0.99 | -27.24 | 0.96 | -21.88 | 0.63 | -13.82 | 0.97 | 14.9 | 0.97 | 31.63 | 1 | 6.42 | 1 | 15.4 |
| 36 | 0.9 | 33.26 | 0.75 | -13.22 | 0.96 | -22.25 | Insufficient data | | 0.67 | -6.83 | 0.94 | -10 | 0.91 | 33.16 | 0.98 | 26.31 | 0.98 | 28.11 | Insufficient data | |
| 37 | 0.99 | 4.43 | 0.99 | -19.04 | 0.98 | -19.04 | 0.98 | -19.04 | 0.98 | -17.37 | 0.94 | -33.26 | 0.92 | 13.43 | Insufficient data | | | | | |
| 38 | 0.88 | 30.05 | 0.65 | -11.62 | 0.96 | -23.28 | 0.96 | -23.28 | 0.87 | 3.96 | 0.94 | -43.54 | 0.94 | 17.63 | 0.95 | 16.92 | 0.99 | 18.77 | 1 | 14.79 |
| 39 | 0.87 | 30.89 | 0.66 | -11.81 | 0.97 | -25.36 | 0.97 | -25.36 | 0.84 | 4.52 | 0.94 | -44.54 | 0.8 | 30.89 | 0.8 | 30.89 | 1 | 4.6 | 1 | 7.4 |
| 40 | 0.77 | 33.1 | 0.49 | 3.2 | 0.9 | -4.81 | 0.9 | -4.81 | 0.91 | -4.8 | 0.94 | -42.81 | 0.77 | 33.1 | 0.77 | 33.1 | 0.99 | 17.96 | 1 | 15.92 |

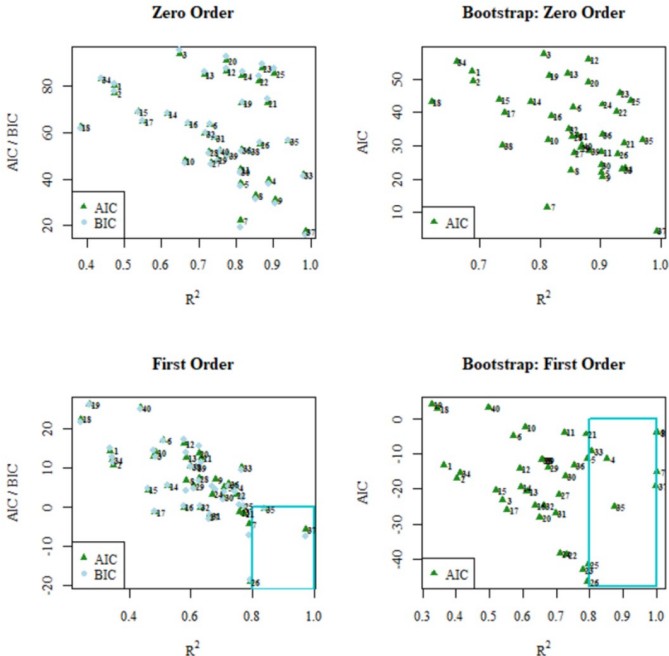

**Fig 5. AIC / BIC vs. R² plots obtained from linear regression fitting (left) and AIC versus R2 obtained from Bootstrap fitting (right) for the zero-order (top) and first-order (bottom) mathematical models.**

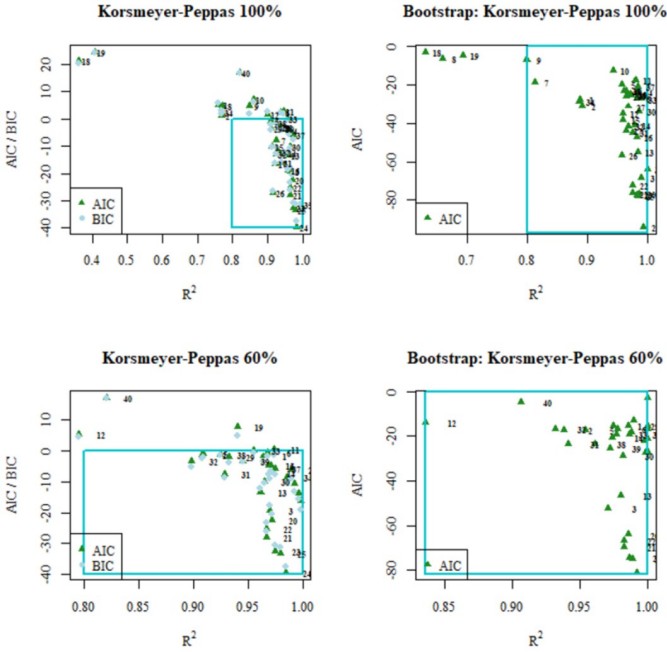

**Fig 6. AIC / BIC vs R² plots obtained from linear regression fitting (left) and AIC vs R2 obtained from Bootstrap fitting (right) for the mathematical models of Korsmeyer-Peppas fitted to the entire release curve (top) and Korsmeyer-Peppas fitted to the first 60% of the release curve (bottom).**

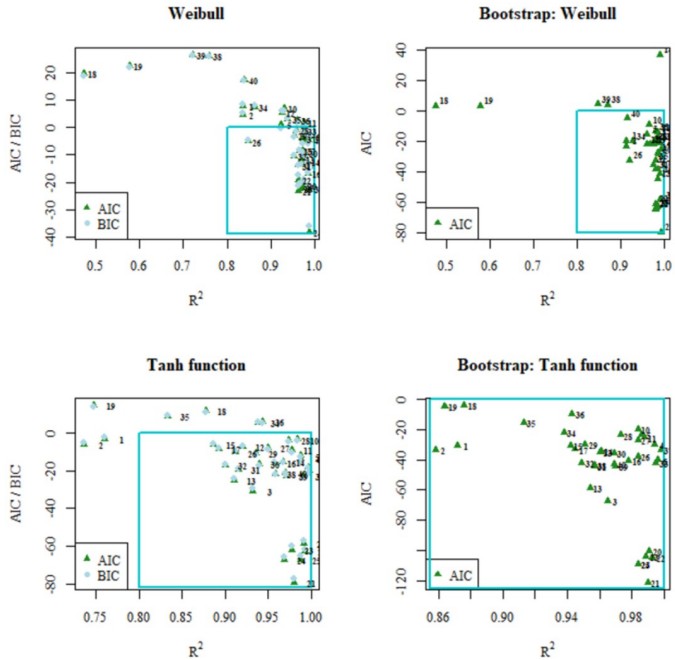

**Fig 7. AIC / BIC vs R$^2$ plots obtained from linear regression fitting (left) and AIC vs R2 obtained from Bootstrap fitting (right) for the mathematical models of Weibull (top) and Hyperbolic Tangent Function (bottom).**

logarithmic form to dampen the abrupt changes in drug concentration rate over time as they pass from one phase to another, has allowed this model to be successfully applied in several release studies [58, 131, 135, 137, 138]. In this study, the Weibull equation was able to fit 12 data sets adequately according to the R$^2$ and R$^2$ adjusted, however, when studying the value of AIC and BIC, the model failed to have the best fit in any of the sets, probably because as an empirical model, its predictive power is poor [58, 62, 137]. Weibull has two geometric parameters. The first one is the scale factor "α" which is part of the intercept with the vertical axis, expressed as "ln($\alpha$)". The second one is the slope of this equation, usually called "β", is a shape factor of the release curve. Thus, if β = 1, the curve maintains an exponential shape, if β<1 it is considered parabolic and in the case of having a β>1, the shape of the curve is sigmoidal [139]. When analyzing this parameter in the collected data sets (S6 File), a tendency towards a parabolic shape could be observed by obtaining an average of β = 0.71. Sets 8, 9, 18,19, 33, 38 and 39 were the only ones with a sigmoidal curve (β>1) [78]. In order to relate the Weibull model to the predominant transport mechanism in drug release, relationships between the Weibull parameter "β" and the Korsmeyer-Peppas parameter "*n*" have been proposed [140]. Following this theory, Fickian diffusion would be demonstrated when β≤0.75, anomalous transport (diffusion and swelling) when β is between 0.75 and 1, case II type transport when β = 1 and case II type transport which is the combination of phenomena such as diffusion, erosion and macromolecular ratio of polymer chains for the case of cylindrical tablets with β>1 [141, 142]. Doing this analysis with "n" and "β" obtained in this study (S6 File) we obtain a value of "β" to check Fickian diffusion of 0.75, while the transport type case II is explained by β = 1.45. The differences in the ranges could be explained by the difference in the geometry of the DDS [143]. This analysis together with the Korsmeyer-Peppas "n" parameter tests the influence of Fickian diffusion on drug release from PLGA nanoparticles [135].

The hyperbolic tangent function model was the one with the best fit to the experimental data analyzed. This mathematical expression was proposed by Eltayeb et al. in 2016 and is derived from the Korsmeyer-Peppas model, considering diffusion as the main transport phenomenon involved in drug release from nanoparticles [78]. This equation uses the hyperbolic tangent of time as a way to mitigate the abrupt effect of the released drug concentration when its phase changes, similar to the effect of the logarithm in the Weibull model. Unlike Korsmeyer-Peppas, this model can be fitted to the complete release profile and not just to its first 60%. Besides, it is applicable to homogeneous matrices, where the ratio of drug and polymer is equal [73, 76]. From the data sets analyzed, 18 sets were fitted to the hyperbolic tangent model by $R^2$ and $R^2_a$ and 27 sets considering the AIC and BIC, indicating the model's ability to describe and predict the drug release profile from PLGA nanoparticles synthesized by nanoprecipitation. Being a relatively new model, the physicochemical significance of its parameters is still not fully understood, however, its authors consider that they are related to particle size and diffusion constant [76]. Furthermore, to our knowledge, this model has not been used by other research groups, however, through the analysis performed in this project it is demonstrated that it can be a useful and statistically proven tool to describe release profiles.

As for the four polynomial equations, the importance of the statistical tests to reject the models was evident. Although they showed a good fit measured by $R^2$ and $R^2_a$, the increase of these does not indicate a good fit of the model since they tend to increase with the number of variables analyzed in the equation, as shown in Fig 2 [143]. Also, when working with polynomial models, we always try to minimize the number of parameters used in the model since, if the number of parameters is equal to n-1, where n is the number of points to be analyzed, we would obtain an error of fit equal to zero, which is described as an overfitting [144]. Equally, the values of AIC and BIC were more positive compared to the mathematical models. Since AIC and BIC are a measure of goodness-of-fit conditional on the number of parameters used in the model, this result is congruent, and indicates the inability of the expressions to predict the future behavior of the system [131, 145]. Pourtalebi Jahromi and collaborators (2020) used the second degree polynomial equation in their analysis of the release kinetics from nanoparticles, obtaining results similar to those of this study [58]. Nevertheless, no literature was found in which the third-, fourth- or fifth-degree equations were used as a mathematical model in drug release.

On the other hand, the number of observations per set limited in some cases the application of the mathematical models and their statistical tests, especially in sets 7, 8 and 9. In these sets the authors reported the release of 100% of the encapsulated drug in 3 observations. According to Jenkins and Quintana-Ascencia (2020), having a minimum of 8 observations would allow significant statistical conclusions to be reached; however, to increase reproducibility, it is recommended that a minimum of 25 observations be used [146]. From the data sets collected, 50% of them have 8 or more observations, with sets number 20–25 having the most number of observations (14 each set).

When Bootstrap resampling was performed, it was possible to determine the model that best fit the data sets 7, 8 and 9. It proves that this technique can be a powerful tool when there is a limited number of observations. The results with this method were similar to the raw data, obtaining that 70% of the data sets fit the Hyperbolic Tangent model.

## Conclusions

This study demonstrated that, although they are tools that help to understand drug release dynamics, there is no general empirical/semiempirical mathematical expression that can describe, in all cases, the release profile of drugs encapsulated in PLGA nanoparticles

synthesized by nanoprecipitation methods. In addition, it was found that although $R^2$ and $R^2_a$ are the most commonly used criteria to determine whether or not a model fits the data obtained in an experiment, they are not the most appropriate. It was also shown that the Akaike and Bayesian criteria can better reflect the fit of a model since the results are not influenced by the inclusion of new terms or an increase in the complexity of the equation. It was also revealed that the number of observations per set is a limiting factor for the application of different models and the subsequent statistical analysis. Therefore, the Bootstrap resampling technique becomes a very useful technique to solve this drawback. In the specific case of this study, 50% of the sets studied do not meet the requirement of at least 8 observations for the construction of the release curve, therefore these results are not very reproducible and have low statistical significance.

Furthermore, the analysis employed in this project provided significant statistical evidence to consider the Hyperbolic Tangent Function model as the most adequate and general model to describe the drug release kinetics. This model, unlike the mathematical expression of Korsmeyer-Peppas, could be adjusted to the complete release curve. Since it has kinetic parameters, it acquires greater predictive power than the Weibull model. However, a more exhaustive study of this model is required in order to understand the chemistry, physics, and biology behind it.

## Supporting information

**S1 File. Equations of second to fifth order polynomials and Equations for the mathematical model acceptance criteria.**
(DOCX)

**S2 File. Release kinetic data.** Forty sets of cumulative drug release data obtained from different scientific articles published in scientific journals since 2016 are listed below. It is included, if available, the model that best fits each set according to the authors.
(DOCX)

**S3 File. R programming.** This appendix shows the programming models in the statistical program R. For this purpose, set number 33 [109] and its fit to the Korsmeyer-Peppas model was taken as an example.
(DOCX)

**S4 File. Results of the fit and the acceptance criteria for each studied mathematical model and polynomial.**
(DOCX)

**S5 File. Bootstrapping fitted regression for each studied mathematical model and polynomial.**
(DOCX)

**S6 File. Linear regression of $\beta$ vs. $n$.** In order to give significance to the Weibull method, a linear regression was performed between the Korsmeyer-Peppas n parameter and the Weibull beta parameter.
(DOCX)

## Acknowledgments

The authors thank to ESPE-INNOVATIVA and Pontificia Universidad Católica del Ecuador (PUCE) for supporting the publication of this article.

## Author Contributions

**Conceptualization:** Karla Vizuete, Alexis Debut.

**Data curation:** Nathaly S. Heredia, Karla Vizuete.

**Formal analysis:** Nathaly S. Heredia, Karla Vizuete.

**Methodology:** Nathaly S. Heredia, Karla Vizuete, Marco Flores-Calero, Katherine Pazmiño V., Alexis Debut.

**Software:** Nathaly S. Heredia.

**Supervision:** Karla Vizuete, Marco Flores-Calero, Fernanda Pilaquinga, Brajesh Kumar.

**Validation:** Karla Vizuete, Katherine Pazmiño V., Fernanda Pilaquinga, Brajesh Kumar, Alexis Debut.

**Writing – original draft:** Nathaly S. Heredia, Karla Vizuete, Alexis Debut.

**Writing – review & editing:** Karla Vizuete, Marco Flores-Calero, Katherine Pazmiño V., Fernanda Pilaquinga, Brajesh Kumar, Alexis Debut.

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
