## [Decision Letter · Decision Letter 0]

2 Dec 2021

PONE-D-21-28541Mathematical modeling of release kinetics from Poly(lactic-co-glycolic acid) nanoparticles synthesized using nanoprecipitation methods: a statistical analysisPLOS ONE

Dear Dr. KS Vizuete,

Apologies for any inconvenience caused due to the delay in the review process. 

Thank you for submitting your manuscript to PLOS ONE. After careful consideration, we feel that it has merit but does not fully meet PLOS ONE’s publication criteria as it stands now. Therefore, we invite you to submit a revised version of the manuscript that addresses the points raised during the review process.

We look forward to receiving your revised manuscript.

Kind regards,

Pratheep K. Annamalai

Academic Editor

PLOS ONE

Journal Requirements:

“This study was funded by Universidad de las Fuerzas Armadas ESPE (website: www.espe.edu.ec) under the project 2020-PIC-010-CTE, Pontificia Universidad Católica del Ecuador (website: www.puce.edu.ec), and ESPE-INNOVATIVA EP (website: www.espe-innovativa.edu.ec). The funders had no role in study design, data collection and analysis, decision to publish, or preparation of the manuscript. There was no additional external funding received for this study.”

“The authors thank to Universidad de las Fuerzas Armadas ESPE, ESPE-INNOVATIVA and Pontificia Universidad Católica del Ecuador (PUCE) for supporting the publication of this article which has been developed under the project 2020-PIC-010-CTE”

“This study was funded by Universidad de las Fuerzas Armadas ESPE (website: www.espe.edu.ec) under the project 2020-PIC-010-CTE, Pontificia Universidad Católica del Ecuador (website: www.puce.edu.ec), and ESPE-INNOVATIVA EP (website: www.espe-innovativa.edu.ec). The funders had no role in study design, data collection and analysis, decision to publish, or preparation of the manuscript. There was no additional external funding received for this study.”

Additional Editor Comments:

This study reports a statistical comparative analysis of the mathematical models for drug delivery systems based on biodegradable polymeric nanoparticles that were reported in last 5 years. This study is worth publishing, after minor revisions suggested by the reviewers.

A suggestion to consider for the title to cover the work and enhance the readership.

"A comparative statistical analysis of the mathematical models for drug delivery systems based on poly(lactic-co-glycolic acid) nanoparticles"

or

"Mathematical modeling of release kinetics from biodegradable polymeric nanoparticles prepared through nanoprecipitation methods: A comparative statistical analysis"

or

"Comparative statistical analysis of the release kinetics models for nanoprecipitated drug delivery systems based on poly(lactic-co-glycolic acid)"

Reviewers' comments:

Reviewer's Responses to Questions

**Comments to the Author**

1. Is the manuscript technically sound, and do the data support the conclusions?

Reviewer #1: Partly

Reviewer #2: Yes

2. Has the statistical analysis been performed appropriately and rigorously? 

Reviewer #1: Yes

Reviewer #2: Yes

3. Have the authors made all data underlying the findings in their manuscript fully available?

Reviewer #1: Yes

Reviewer #2: Yes

4. Is the manuscript presented in an intelligible fashion and written in standard English?

Reviewer #1: Yes

Reviewer #2: Yes

5. Review Comments to the Author

Reviewer #1: Authors deal with the statistical comparison of mathematical models used in order to find which of them best describes the drug release profile from PLGA particles synthesized by nanoprecipitation method. I think the manuscript is very well written overall, very straightforward to follow and understand

I will recommend their publication after few corrections/suggestions as detailed below.

Why authors only consider 40 works and all of them after 2016? Perhaps a clear explanation it would be nice for readers.

I would also suggest to provide extra 1 or 2 self-made schematizations. For instance: i.- Time line regarding the mathematical models used so far to describes the drug release in similar materials. Ii.- a graphical schematization regarding their main results

Few english sentences are confusing. I will suggest to check/correct the whole manuscript by a native

I’ve particularly don’t like the bunch of tables in the manuscript. I would suggest to maintain in the main text the most appealing tables and move the rest to the SI.

Conclusion need to be rewritten, to highlight their main findings.

Reviewer #2: The manuscript applied the appropriate statistical approaches to compare and summarize the commonly used models for the drug release profile from PLGA particles synthesized by nano-precipitation method.

The screening and justification of literature and data are important for the assessment of study outcomes and final conclusion. It will be better to add a flow diagram for the screening procedure and justification/criteria of literature for the systematic research.

6. PLOS authors have the option to publish the peer review history of their article (what does this mean?). If published, this will include your full peer review and any attached files.

Reviewer #1: No

Reviewer #2: No

---

## [Author Response · Author response to Decision Letter 0]

14 Jan 2022

Pratheep K. Annamalai

Academic Editor

PLOS ONE

Re: Resubmision of manuscript reference No. PONE-D-21-28541

Please find attached a revised version of our manuscript originally entitled “Mathematical modeling of release kinetics from Poly(lactic-co-glycolic acid) nanoparticles synthesized using nanoprecipitation methods: a statistical analysis” which we would like to resubmit for publication to Plos One. 

We appreciate the time and effort that you and the reviewers have dedicated to provide your valuable feedback. Your comments and those of the reviewers were highly insightful and enabled us to greatly improve the quality of our manuscript. The authors have carefully considered the comments and tried our best to address every one of them. Our revisions to the text are recorded using Track Changes in Microsoft Word. Our point-by-point responses to your own comments are shown below, and the responses to reviewers’ comments are presented in Response to Reviewers Word document. 

We hope that the revisions in the manuscript and our accompanying responses will be sufficient to make our manuscript suitable for publication in Plos One. 

If any responses are unclear or you wish additional changes, please let us know

We shall look forward to hearing from you at your earliest convenience.

Yours sincerely,

Karla Vizuete, Mtr. 

Universidad de las Fuerzas Armadas ESPE

Av. General Rumiñahui S/N and Ambato 

Sangolquí, 171103, Ecuador 

(+593) 2 398 9400 Ext. 2113 ksvizuete@espe.edu.ec

For a better reading, editor comments are shown in bold blue, while our response is shown in black. 

Journal Requirements:

Response:

We have carefully reviewed the templates and we have changes authors’ affiliations:

Original: 

1Departamento de Ciencias de la Vida y la Agricultura, Universidad de las Fuerzas Armadas ESPE, Sangolquí 171103, Ecuador

2Centro de Nanociencia y Nanotecnología CENCINAT, Universidad de las Fuerzas Armadas ESPE, Sangolquí 171103, Ecuador

3Departamento de Eléctrica, Electrónica y Telecomunicaciones, Universidad de las Fuerzas Armadas ESPE, Sangolquí 171103, Ecuador

4Escuela de Ciencias Químicas, Pontificia Universidad Católica del Ecuador, Quito 170525, Ecuador

5Department of Chemistry, TATA College, Chaibasa 833202, India

* Corresponding author: 

E-mail: ksvizuete@espe.edu.ec

Revised: 

1Departamento de Ciencias de la Vida y la Agricultura, Universidad de las Fuerzas Armadas ESPE, Sangolquí, Pichincha, Ecuador

2Centro de Nanociencia y Nanotecnología CENCINAT, Universidad de las Fuerzas Armadas ESPE, Sangolquí, Pichincha, Ecuador

3Departamenteo de Eléctrica, Electrónica y Telecomunicaciones, Universidad de las Fuerzas Armadas ESPE, Sangolquí, Pichincha, Ecuador

4Escuela de Ciencias Químicas, Pontificia Universidad Católica del Ecuador, Quito, Pichincha, Ecuador

5Department of Chemistry, TATA College, Chaibasa, Jharkhand, India

* Corresponding author: 

E-mail: ksvizuete@espe.edu.ec (KV)

“This study was funded by Universidad de las Fuerzas Armadas ESPE (website: www.espe.edu.ec) under the project 2020-PIC-010-CTE, Pontificia Universidad Católica del Ecuador (website: www.puce.edu.ec), and ESPE-INNOVATIVA EP (website: www.espe-innovativa.edu.ec). The funders had no role in study design, data collection and analysis, decision to publish, or preparation of the manuscript. There was no additional external funding received for this study.”

Response:

In response to the requirements of items 2 and 3, the financial disclosure was changed to:

“This study was financially funded by Universidad de las Fuerzas Armadas ESPE (website: www.espe.edu.ec) under the project 2020-PIC-010-CTE led by A.D., Pontificia Universidad Católica del Ecuador (website: www.puce.edu.ec), and ESPE-INNOVATIVA EP (website: www.espe-innovativa.edu.ec). K.P.V.'s salary was covered by ESPE-INNOVATIVA. The funders had no role in study design, data collection and analysis, decision to publish, or preparation of the manuscript. There was no additional external funding received for this study.”

“The authors thank to Universidad de las Fuerzas Armadas ESPE, ESPE-INNOVATIVA and Pontificia Universidad Católica del Ecuador (PUCE) for supporting the publication of this article which has been developed under the project 2020-PIC-010-CTE”

“This study was funded by Universidad de las Fuerzas Armadas ESPE (website: www.espe.edu.ec) under the project 2020-PIC-010-CTE, Pontificia Universidad Católica del Ecuador (website: www.puce.edu.ec), and ESPE-INNOVATIVA EP (website: www.espe-innovativa.edu.ec). The funders had no role in study design, data collection and analysis, decision to publish, or preparation of the manuscript. There was no additional external funding received for this study.”

Response:

Since no other person or institution, other than the authors and those mentioned in the financial disclosure, collaborated in the development and writing of the manuscript, we have eliminated this section from the draft.

Response:

All data used in this study are reported in the body of the manuscript or in the supplementary information. Specifically, the data sets extracted from the literature are detailed in the supplementary information as well as the statistical analysis procedure and results for each data set. The manuscript presents information on the particles obtained in each data set and summarizes the results of the statistical analysis.

Response:

We apologize for the error. We are not going to provide repository information for our data at acceptance. We have already submitted all the information and data necessary to support the results presented in the manuscript.

Additional Editor Comments:

This study reports a statistical comparative analysis of the mathematical models for drug delivery systems based on biodegradable polymeric nanoparticles that were reported in last 5 years. This study is worth publishing, after minor revisions suggested by the reviewers.

A suggestion to consider for the title to cover the work and enhance the readership.

"A comparative statistical analysis of the mathematical models for drug delivery systems based on poly(lactic-co-glycolic acid) nanoparticles"

or

"Mathematical modeling of release kinetics from biodegradable polymeric nanoparticles prepared through nanoprecipitation methods: A comparative statistical analysis"

or

"Comparative statistical analysis of the release kinetics models for nanoprecipitated drug delivery systems based on poly(lactic-co-glycolic acid)"

Response:

We are most grateful for the time the editor spent on providing title suggestions. We discussed it a lot within our team and selected the title: "Comparative statistical analysis of the release kinetics models for nanoprecipitated drug delivery systems based on poly(lactic-co-glycolic acid)" as the option that best fits our scope. Thanks again for this invaluable suggestion. The title has been changed in the manuscript as well as in supporting information. 

5. Review Comments to the Author

Responses to the comments of Reviewer #1

Thank you for forwarding these helpful review reports. We have tried to address REVIEWER #1 suggestions as well as possible and specified in detail below.

Reviewer #1: Authors deal with the statistical comparison of mathematical models used in order to find which of them best describes the drug release profile from PLGA particles synthesized by nanoprecipitation method. I think the manuscript is very well written overall, very straightforward to follow and understand

I will recommend their publication after few corrections/suggestions as detailed below.

1. Why authors only consider 40 works and all of them after 2016? Perhaps a clear explanation it would be nice for readers.

Response:

The authors thank this valuable comment. We considered 40 datasets extracted from scientific literature since 2016. This is due to three main reasons listed below. 

1. Using the nanoprecipitation method there are hundreds of scientific papers published so far, therefore, to limit the study, the papers from the last 5 years, i.e. since 2016, were selected. 

2. In 2016, to the best of our knowledge, a paper referring to the hyperbolic tangent function model was published for the first time (Eltayeb, Stride, Edirisinghe, & Harker, 2016).

3. The authors considered the time of 5 years as sufficient for other authors to consider and use the hyperbolic tangent model and thus collect representative data and results. 

For a better understanding of the readers an explanation of this was added in lines 170-172 of the manuscript where it is expressed:

To limit the study, the authors selected scientific articles from 2016. Furthermore, to the best of our knowledge, this was the year in which the mathematical model of hyperbolic tangent function was applied for the first time to evaluate the release profile of core-shell lipid nanoparticles [77].

[77] Eltayeb, M., Stride, E., Edirisinghe, M., & Harker, A. (2016). Electrosprayed nanoparticle delivery system for controlled release. Materials Science and Engineering C, 66, 138–146. https://doi.org/10.1016/j.msec.2016.04.001

2. I would also suggest to provide extra 1 or 2 self-made schematizations. For instance: i.- Time line regarding the mathematical models used so far to describes the drug release in similar materials. ii.- a graphical schematization regarding their main results.

Response:

Accordingly to this comment, we have added the following schematization to address point i) as figure #2.

Here the authors show a timeline of the development of the models used in the present study. 

In the manuscript, line 200-203 introduce the reader to the aforementioned timeline:

3. Few English sentences are confusing. I will suggest to check/correct the whole manuscript by a native

Response:

Thank you for your suggestion, whole manuscript has been revised by a Native American. All changes are recorded using Track Changes in Microsoft Word 

4. I’ve particularly don’t like the bunch of tables in the manuscript. I would suggest to maintain in the main text the most appealing tables and move the rest to the SI.

Response:

Thank you for your suggestion. We have moved Table 1 and 2 to SI. These tables were selected because they are not main results. Table 1 corresponds to standard equations and Table 2 to scientifical well-known definitions and general equations. 

It should be noted that the numbers of the figures and tables in the manuscript and in the SI have changed, these have been updated and also their citations within the text.

5. Conclusion need to be rewritten, to highlight their main findings.

Response:

Thank you for pointing this out. Conclusion has been rewritten as follow (lines 524-541):

This study demonstrated that, although they are tools that help to understand drug release dynamics, there is no general empirical/semiempirical mathematical expression that can describe, in all cases, the release profile of drugs encapsulated in PLGA nanoparticles synthesized by nanoprecipitation methods. In addition, it was found that although R2 and R2a are the most commonly used criteria to determine whether or not a model fits the data obtained in an experiment, they are not the most appropriate. It was also shown that the Akaike and Bayesian criteria can better reflect the fit of a model since the results are not influenced by the inclusion of new terms or an increase in the complexity of the equation. It was also revealed that the number of observations per set is a limiting factor for the application of different models and the subsequent statistical analysis. Therefore, the Bootstrap resampling technique becomes a very useful technique to solve this drawback. In the specific case of this study, 50% of the sets studied do not meet the requirement of at least 8 observations for the construction of the release curve, therefore these results are not very reproducible and have low statistical significance. 

Furthermore, the analysis employed in this project provided significant statistical evidence to consider the Hyperbolic Tangent Function model as the most adequate and general model to describe the drug release kinetics. This model, unlike the mathematical expression of Korsmeyer-Peppas, could be adjusted to the complete release curve. Since it has kinetic parameters, it acquires greater predictive power than the Weibull model. However, a more exhaustive study of this model is required in order to understand the chemistry, physics, and biology behind it.

Responses to the comments of Reviewer #2

Reviewer #2: The manuscript applied the appropriate statistical approaches to compare and summarize the commonly used models for the drug release profile from PLGA particles synthesized by nano-precipitation method.

The screening and justification of literature and data are important for the assessment of study outcomes and final conclusion. It will be better to add a flow diagram for the screening procedure and justification/criteria of literature for the systematic research.

Response:

We think this is an excellent suggestion, therefore, we have added Fig. 1 and its citation in the manuscript.

For a better understanding of the readers an explanation of the selection criteria was added in lines 170-172 of the manuscript where it is expressed:

To limit the study, the authors selected scientific articles from 2016. Furthermore, to the best of our knowledge, this was the year in which the mathematical model of hyperbolic tangent function was applied for the first time to evaluate the release profile of core-shell lipid nanoparticles [77].

[77] Eltayeb, M., Stride, E., Edirisinghe, M., & Harker, A. (2016). Electrosprayed nanoparticle delivery system for controlled release. Materials Science and Engineering C, 66, 138–146. https://doi.org/10.1016/j.msec.2016.04.001

---

## [Decision Letter · Decision Letter 1]

18 Feb 2022

Comparative statistical analysis of the release kinetics models for nanoprecipitated drug delivery systems based on poly(lactic-co-glycolic acid)

PONE-D-21-28541R1

Dear Dr. Vizuete,

Thank you for your patience. After evaluating the reviewers' comments, the responses and revised submission, we’re pleased to inform you that your manuscript has been judged scientifically suitable for publication and will be formally accepted for publication once it meets all outstanding technical requirements.

Kind regards,

Pratheep K. Annamalai

Academic Editor

PLOS ONE

Additional Editor Comments (optional):

The Authors are appreciated for addressing the comments.

Reviewers' comments:

Reviewer's Responses to Questions

**Comments to the Author**

1. If the authors have adequately addressed your comments raised in a previous round of review and you feel that this manuscript is now acceptable for publication, you may indicate that here to bypass the “Comments to the Author” section, enter your conflict of interest statement in the “Confidential to Editor” section, and submit your "Accept" recommendation.

Reviewer #1: All comments have been addressed

Reviewer #2: All comments have been addressed

2. Is the manuscript technically sound, and do the data support the conclusions?

Reviewer #1: Partly

Reviewer #2: Yes

3. Has the statistical analysis been performed appropriately and rigorously? 

Reviewer #1: Yes

Reviewer #2: Yes

4. Have the authors made all data underlying the findings in their manuscript fully available?

Reviewer #1: Yes

Reviewer #2: Yes

5. Is the manuscript presented in an intelligible fashion and written in standard English?

Reviewer #1: Yes

Reviewer #2: Yes

6. Review Comments to the Author

Reviewer #1: (No Response)

Reviewer #2: The manuscript applied the appropriate statistical approaches to compare and summarize the commonly used models for the

drug release profile from PLGA particles synthesized by nano-precipitation method.

The suggestion to add a flowchart for screening literature and data was addressed by the author.

7. PLOS authors have the option to publish the peer review history of their article (what does this mean?). If published, this will include your full peer review and any attached files.

Reviewer #1: No

Reviewer #2: No

---

## [Editor Report · Acceptance letter]

24 Feb 2022

PONE-D-21-28541R1 

Comparative statistical analysis of the release kinetics models for nanoprecipitated drug delivery systems based on poly(lactic-co-glycolic acid) 

Dear Dr. Vizuete:

I'm pleased to inform you that your manuscript has been deemed suitable for publication in PLOS ONE. Congratulations! Your manuscript is now with our production department. 

Kind regards, 

on behalf of

Dr. Pratheep K. Annamalai 

Academic Editor

PLOS ONE